# DEM Simulation of Laboratory-Scale Jaw Crushing of a Gold-Bearing Ore Using a Particle Replacement Model

**Gabriel Kamilo Barrios [1,\*]**, **Narcés Jiménez-Herrera [1,\*]**, **Silvia Natalia Fuentes-Torres [1]** and **Luís Marcelo Tavares [2]**

1   Colombian Geological Survey (CGS), Cali 760001, Colombia; sfuentes@sgc.gov.co
2   Department of Metallurgical and Material Engineering, Laboratory of Mineral Technology, Universidade Federal do Rio de Janeiro, Rio de Janeiro CEP 21941-972, Brazil; tavares@metalmat.ufrj.br
\*   Correspondence: gkpbarrios@metalmat.ufrj.br (G.K.B.); njimenez@metalmat.ufrj.br (N.J.-H.); Tel.: +(57)-312-622-5883 (G.K.B.)

**Abstract:** The Discrete Element Method (DEM) is a numerical method that is able to simulate the mechanical behavior of bulk solids flow using spheres or polyhedral elements, offering a powerful tool for equipment design and optimization through modeling and simulation. The present work uses a Particle Replacement Model (PRM) embedded in the software EDEM® to model and simulate operation of a laboratory-scale jaw crusher. The PRM was calibrated using data from single particle slow compression tests, whereas simulations of the jaw crusher were validated on the basis of experiments, with very good agreement. DEM simulations described the performance of the crusher in terms of throughput, product size distribution, compressive force on the jaws surface, reduction ratio, and energy consumption as a function of closed side setting and frequency.

**Keywords:** crushing; jaw crusher; Discrete Element Method; Particle Replacement Model; comminution; simulation; modeling; primary crushing; particle breakage

## 1. Introduction

Jaw crushers are widely used in the primary crushing stage and, sometimes, even in the secondary for many applications, including the processing of metallic, non-metallic, energetic, and industrial minerals, as well as in the processing of construction and demolition waste.

Despite the technology being over a century old [1], the original designs of the jaw crushers have been maintained nearly unchanged, taking advantage of the simplicity of their structure and mechanical operation. These features result in ease in manufacturing, repairing, dissembling and low capital cost in comparison to other types of crushers [2].

This machine is composed of two metallic plates forming a V-shape. One of them is fixed while the other swings, moving due to action of an eccentric shaft connected to a motor. When in operation, the ore is fed to the top opening and travels down along the chamber. On its path, the ore is crushed in successive cycles of application of stress, primarily compression, which are applied when the moving plate approaches the fixed plate. In the return movement of the moving plate, the ore particles slip down the chamber until they are stressed once again in the following cycle. The process continues until particles reach a size that is finer than the bottom opening so, in that moment, the ore particles drop out of the crushing chamber [3].

Jaw crushers are robust, being an attractive option in operations where the feed has a coarse top size and a moderate reduction ratio without fines is required [2]. Their performance, in terms of

capacity or throughput, power and energy consumption, depends on material properties, equipment design and operating parameters. On one hand, the material characteristics are given by density, hardness, bulk density, particle size distribution, particle top size and particle strength (toughness) and crushability of the feed material [4]. The design parameters of the crusher include the size of the top opening, the set, the volume of the crushing chamber, and the type of jaw surface, which may be smooth or corrugated [5]. The equipment operational parameters include the frequency and amplitude of the movable jaw stroke, the feed rate, the closed side setting (CSS) of the discharge opening, among others [6,7]. It is worth mentioning that the material that is used to line the jaws must be hard and tough in order to endure impact and wear during operation. Some aspects of the crusher materials and a failure analysis of a jaw crusher have been investigated by Olawale and Ibitoye [8].

Different mathematical models have been developed to describe the performance of the jaw crusher. The first generation models were based on empirical expressions to predict capacity [9,10] and energy consumption [10–12]. Later, more robust mathematical models were proposed, based on the population balance model, to represent more details of the machine performance, including the full product size distribution [13]. More recently, a model that describes the kinematics of the equipment to predict flow, capacity, power, among others, has been proposed [14].

Over the last few decades mechanistic approaches that rely on the Discrete Element Method have shown great value in the description of the performance of different types of crushers. Fusheng et al. [15] and Legendre and Zevenhoven [16] performed DEM simulations of size reduction of a single particle in a jaw crusher, in which the bonded particle model (BPM) was used to describe particle breakage. Particle breakage models in DEM usually require significant computational effort and their application in systems with multiple particles, such as those found in crushers operating in industry, may be complex [17]. Among the particle breakage model approaches compared in a recent review by Jimenez-Herrera et al. [16], PRM using spheres was identified as the one with the lowest computational cost, making it attractive to simulate the machine operation in which the feed is constituted by a stream of particles. Indeed, a successful application of PRM has been demonstrated to selected compression crushers, including a jaw crusher [18].

The present work describes the modeling and simulation of a laboratory-scale jaw crusher using DEM with the particle replacement model embedded to describe product size distribution, throughput and crusher power. The PRM has been implemented as a modification of the Hertz-Mindlin contact model, through which each spherical mother particle is replaced by a distribution of daughter spherical particles every time the mother particle is subjected to a force that surpasses a maximum set value. A comprehensive description of the model used is presented elsewhere [19]. DEM simulations were validated on the basis of experiments in a laboratory jaw crusher in the size reduction of a gold ore. Additionally, a sensitive analysis of the simulation model was carried out to investigate the effects of the closed side setting (CSS) and frequency on capacity, power, compressive force and reduction ratio.

## 2. Materials and Methods

### 2.1. Materials

The material used in this study is a gold-bearing ore from the San José mine in Íquira, Huila region, in Colombia. The ore is an intrusive igneous rock from the Ibagué batholith, with phaneritic texture, coarse-to-medium grain sizes, intermediate felsic composition, being predominantly composed of granodiorites. Table 1 presents the mineralogical composition of the Íquira gold ore determined by optical microscopy, which shows that it is composed mainly of quartz and feldspar as main gangue minerals, pyrite, carbonates with a smaller percentage of other metallic minerals such as galena, hematite and arsenopyrite [20].

**Table 1.** Mineral composition of the gold ore from Íquira-Huila

| Mineral | Percent (%) |
| --- | --- |
| Gangue minerals (quartz and feldspar) | 62 |
| Pyrite | 23 |
| Carbonates | 7 |
| Chalcopyrite | 4 |
| Hematite | 2 |
| Sphalerite | 1 |
| Galena | 1 |

Figure 1 shows a snapshot of the Run of Mine sample collected for testing. The particles were classified into three narrow sizes for testing: 63/53, 31.5/26.5 and 16.0/13.2 mm.

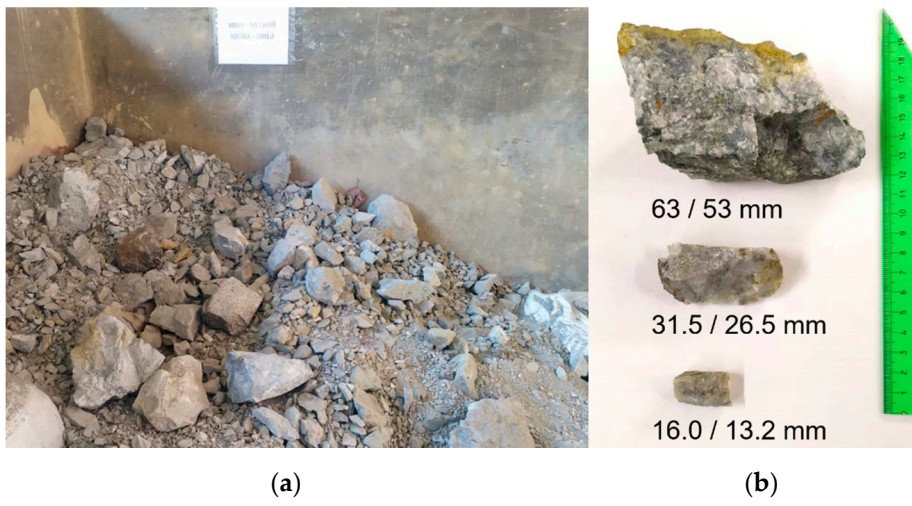

(**a**)　　　　　　　　　　　　　　　　(**b**)

**Figure 1.** Run of Mine sample of the Íquira-Huila gold ore (**a**) and particles classified in narrow sizes (**b**).

### 2.2. Laboratory-Scale Jaw Crushing Tests

Figure 2 shows a snapshot of the Otsuka Iron Works Ltd. laboratory-scale jaw crusher used in the present study, which has the universal design [2]. Table 2 summarizes the main operational parameters of the machine, used in the experimental tests. Each batch crushing test was conducted using about 6 kg of sample with particles contained either in narrow sizes or with a distribution of sizes.

The jaw crusher throughput for each experimental test was measured using an integrating load cell coupled to an Arduino UNO data acquisition device. The net power was calculated from measurements with an amperage multimeter, whereas the product particle size distributions were measured using a $\sqrt{2}$ series of sieves in a laboratory sieve shaker.

**Table 2.** Operational parameters used in the laboratory-scale jaw crushing experimental tests

| Parameters | Units | Value |
| --- | --- | --- |
| Nominal throughput | kg/h | 300 |
| Power | kW | 2.2 |
| Main shaft frequency | rpm | 400 |
| Swing jaw stroke frequency | Hz | 6.0 |
| Feed (top) opening | mm × mm | 140 × 90 |
| Discharge opening | mm × mm | 140 × 7.5 (closed) |
| Swing jaw throw | mm | 10 |

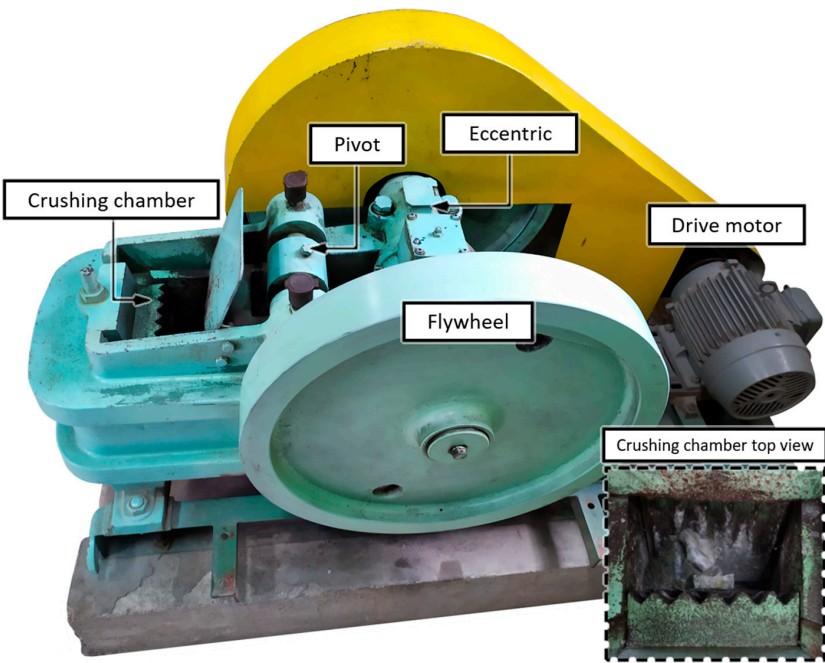

**Figure 2.** Laboratory-scale jaw crusher at the Colombian Geological survey, with the insert showing the top view of the crushing chamber.

Figure 3 shows the jaw crusher scheme indicating the main components of the machine and the operating variables such as the discharge opening and the displacement of the swing jaw.

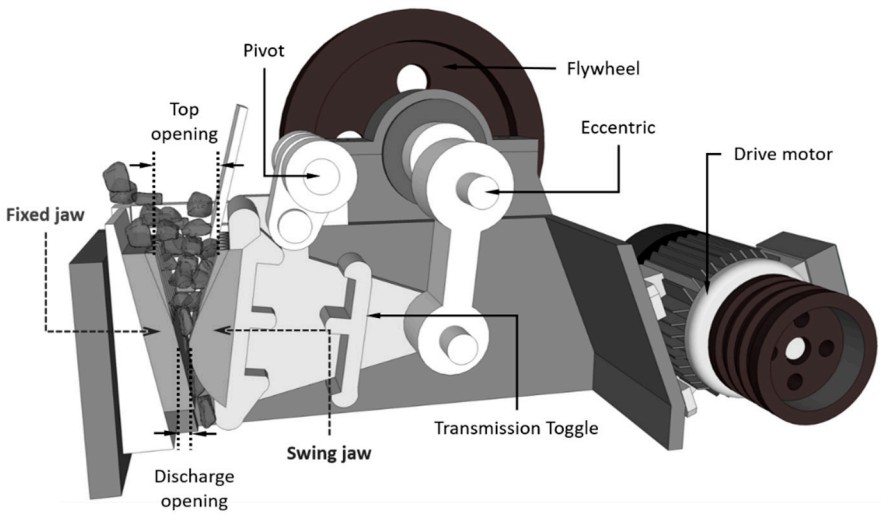

**Figure 3.** Scheme of the jaw crusher showing the main components of the machine.

### 2.3. DEM Particle Replacement Model Parameter Calibration

The particle replacement model (PRM) used in this work may be used to describe fragmentation of individual particles under compression or impact. It consists of instantaneously replacing a spherical particle by progeny fragments every time a critical condition for failure is met. Spheres that make up the progeny fragments are allowed to overlap each other at the instant of replacement and a relaxation factor of the repulsive force was applied to prevent them from explosively repelling each other given the large contact forces involved. The version used in the present work is a custom PRM implemented in the software EDEM®, using an Advanced Programming Interface (API). An extended description of the model is presented in the work of Barrios et al. (2020) [19].

The PRM formulation used in the present work has been successfully used previously in DEM simulations describing breakage of individual and beds of particles by impact [17], as well as the interaction of particles and complex geometries in High-Pressure Grinding Rolls [21].

The calibration of the PRM parameters was conducted from uniaxial compression tests of individual unconfined irregular particles using a universal press manufacture by Maekawa with a capacity of 400 kN, applying a methodology similar to that used by Qian et al. [22]. Displacement during compression was measured using a digital camera, and the force was measured using a load cell with a maximum capacity of 5 kN, coupled to an Arduino UNO data acquisition device connected to a laptop (Figure 4). Each test consisted of compressing 30 particles contained in each narrow size range.

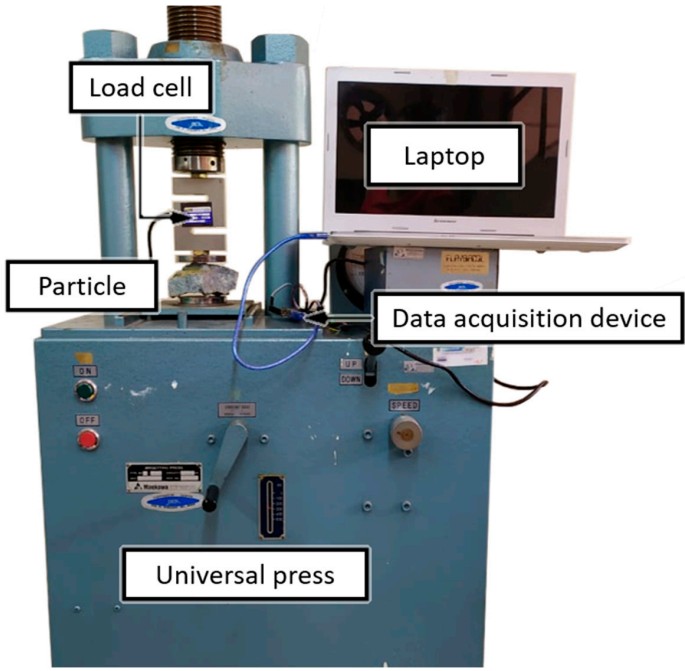

**Figure 4.** Uniaxial compression system used in unconfined testing of particles.

## 2.4. Jaw Crusher DEM Simulations

Tables 3 and 4 show individual and contact parameters of the materials used in the DEM simulations. These DEM parameters were chosen based on the work of Rodriguez et al. [23], and validated on the basis of repose angle results from the EDEM® software database (Generic EDEM Material Model Database—GEMM).

**Table 3.** Individual material parameters used in the DEM simulations

| Parameters | Units | Steel | Ore |
|---|---|---|---|
| Density | kg/m$^3$ | 7800 | 2700 |
| Shear modulus | Pa | $7 \times 10^9$ | $1 \times 10^8$ |
| Poisson's ratio | - | 0.30 | 0.25 |

**Table 4.** Contact parameters used in the DEM simulations

| Parameters | Units | Steel–Ore | Ore–Ore |
|---|---|---|---|
| Coefficient of restitution | - | 0.37 | 0.35 |
| Coefficient of friction | - | 0.20 | 0.34 |
| Coefficient of rolling friction | - | 0.10 | 0.25 |

Material and contact parameters in Tables 3 and 4 were used both in calibration of the PRM parameters and in the DEM simulations of the jaw crusher.

EDEM® 2019 was used in the DEM simulations of the laboratory-scale jaw crusher. The CAD model of the jaw crusher geometry was designed using the software SketchUp (Boulder, CO, USA), based on the Otsuka Iron Works manual, as well as direct measurements of the discharge opening and the jaw wear surfaces. Table 5 shows the ranges of operating variables used in the DEM simulations of the jaw crusher.

The throughput of the simulated jaw crusher was obtained using the "flow sensor" feature of the EDEM® post processing module. The compressive force and the power on the swing and fixed jaws were calculated extracting the force and the torque on each element of the geometry. Finally, the particle size distribution of the product was calculated based on the mass of the particles produced by the PRM.

**Table 5.** Operating conditions adopted in the DEM simulations of the jaw crusher

| Parameter | Units | Range | |
|---|---|---|---|
| Discharge opening | mm | 2.5–7.5–12.5 | |
| Swing jaw stroke frequency | Hz | 0.5–3.0–6.0–9.0 | |
| Feed particle narrow sizes | mm | 63.0/52.0 | 100% |
| | | 31.5/26.5 | 100% |
| | | 16.0/13.2 | 100% |
| Feed particle size distribution | mm | 63.0/52.0 | 33.3% |
| | | 31.5/26.5 | 33.3% |
| | | 16.0/13.2 | 33.3% |

## 3. Results

### 3.1. Calibration of Particle Replacement Model Parameters

Figure 5 shows the comparison between the experimental and the DEM simulation set-up of the single particle compression test, used to calibrate the PRM parameters of mother particle fragmentation and breakage force.

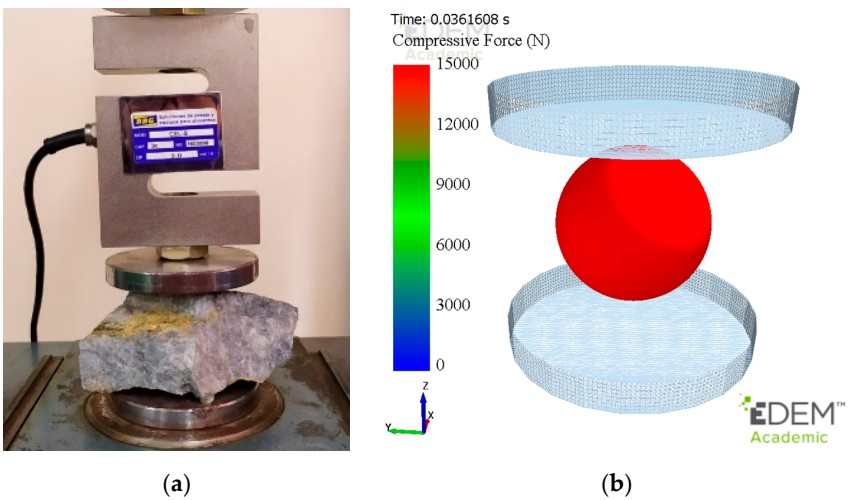

(a)                                                        (b)

**Figure 5.** Experimental (**a**) and DEM simulation (**b**) set up of the single particle uniaxial compression test.

Figure 6a shows a comparison between the experimental and simulated force-deformation profiles of a single particle obtained from a compression test. In the figure, some of the limitations of the PRM, already discussed by Jimenez-Herrera et al. [17], in describing the fine details of the breakage process,

are evident. Figure 6b shows the cumulative distribution of specific fracture energies for each size class. From the experimental distribution the median values of breakage force that is used to calibrate the threshold breakage force parameter of the DEM Particle Replacement Model, above which mother particles break and are replaced by daughter particles, is calculated.

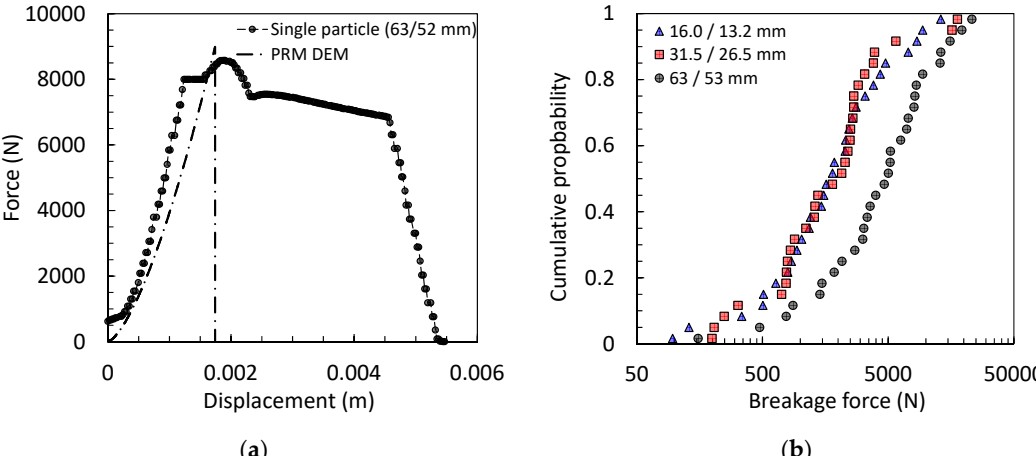

(**a**)             (**b**)

**Figure 6.** Force-deformation profile of a single particle compression test (experiment and simulation) (**a**) and distribution of breakage forces for three size classes (**b**).

Figure 7a shows the experimental product size distributions from the jaw crushing tests conducted using the different feeds, including the three different narrow size classes and the particle size distribution. Figure 7b compares results from modeling and calibration of the DEM PRM spherical daughter particle distribution based on the experimental results of the jaw crushing test using a distributed feed. It was fitted using a primary distribution in which every breakage event results in a generation of daughter particle contained in three size classes: 1 particle with a size ratio equal to 0.595 of the mother particle, 8 particles with size ratio equal to 0.354 of the mother particle and the last with 52 particles with size ratio equal to 0.210 of the mother particle. In addition, the value of the relaxation factor $b_L$ equal to 0.0524 of the particle replacement model [19] was selected, which is responsible for capping the normal force calculated using the overlap of the daughter particles, so as to prevent the appearance of extremely high velocities of the fragments from breakage which would make simulations unrealistic.

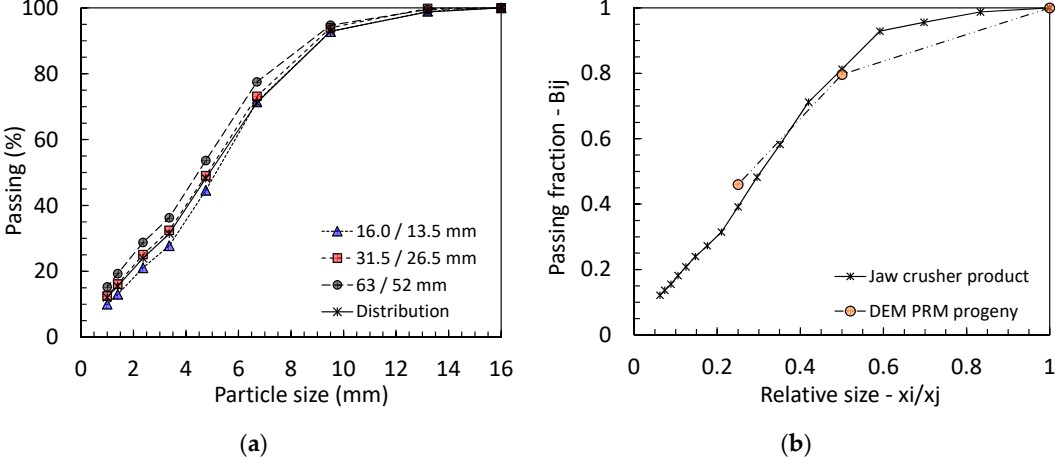

(**a**)             (**b**)

**Figure 7.** Product size distributions from laboratory jaw crushing tests with different feeds (**a**) and DEM PRM daughter spheres distribution modeled (**b**).

### 3.2. DEM Simulations of the Laboratory Jaw Crusher

Figure 8a is a snapshot of the DEM simulation of particles being crushed in the virtual jaw crusher. In this perspective, a view of the breakage of particles, modeled using the Particle Replacement Model, may be observed. On the other hand, Figure 8b shows qualitative results on the velocity profile of the particles along the crushing chamber. In this figure, the higher velocities of the particles appear closer to the discharge, given the greater freedom of the fragments to move downwards in the chamber as they become progressively finer. Such a result shows the capabilities of the DEM model to represent the dynamics of the particles that are expected to appear in real jaw crushers.

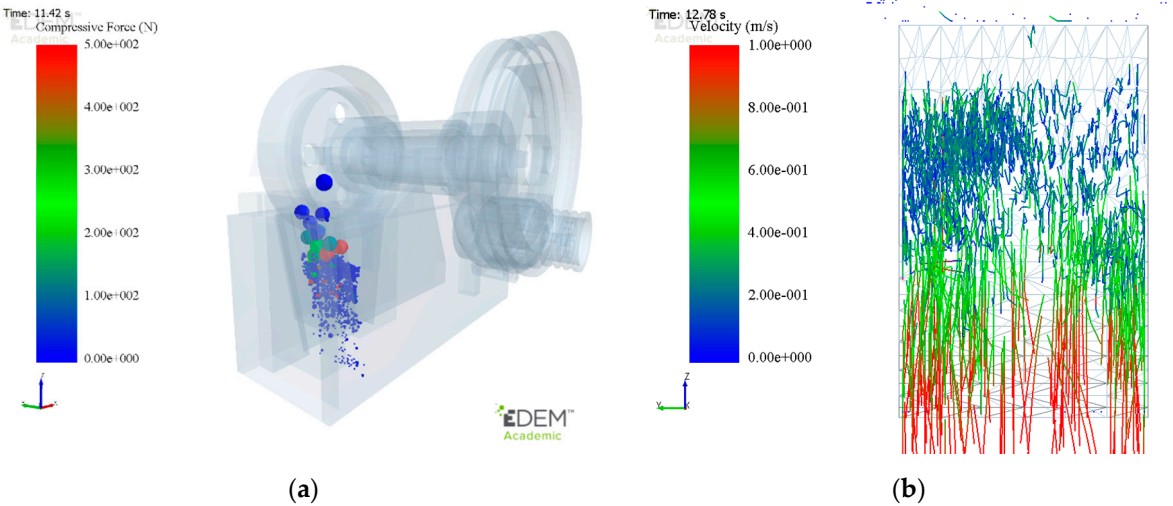

(**a**)                    (**b**)

**Figure 8.** Perspective view of DEM simulation of the laboratory jaw crusher (**a**) and particle velocities profile through the crushing chamber (**b**).

Figure 9 shows a comparison between the experimental and simulated breakage of a single particle contained in the 63/53 mm range inside the jaw crusher chamber. The DEM simulation coupled to the PRM is capable to provide a valid qualitative representation of fragmentation observed in the experiment.

Experimental

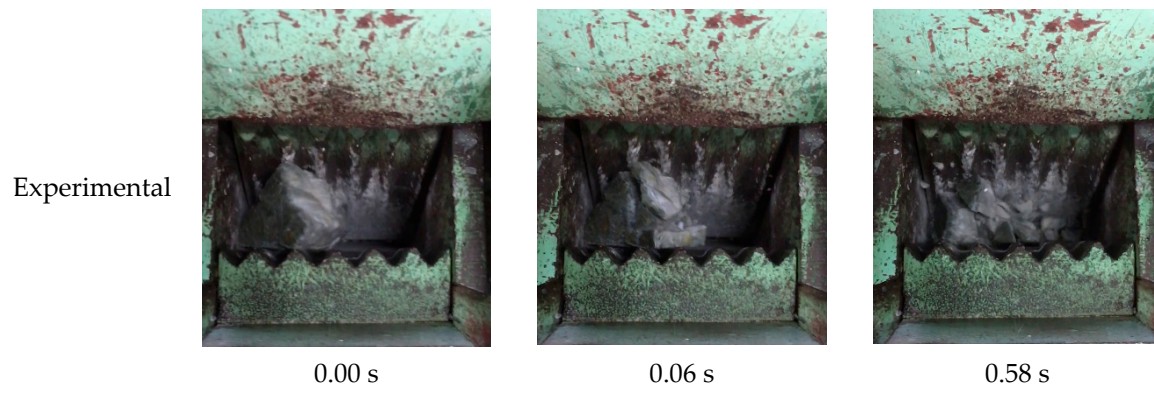

0.00 s                    0.06 s                    0.58 s

**Figure 9.** *Cont.*

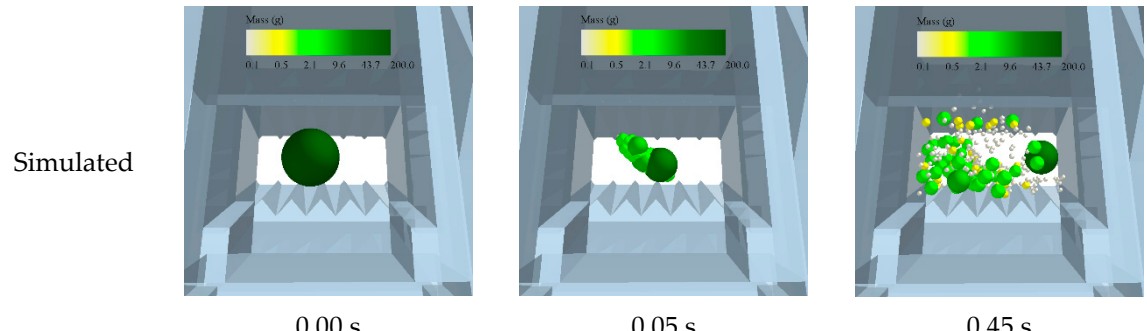

**Figure 9.** Comparison of experimental and simulated breakage of a single 63/52-mm particle in the jaw crusher chamber.

Simulation results are shown in Figure 10 for different feed particle sizes, showing the qualitative results of the particle bed compression within the crushing chamber. The compressive force on the particles is represented by the colored lateral bar. Particle visualization shows the plate zone where the particles are broken according to their sizes and the differences in the stress field exerted on the plates according to particle size. As expected, given the V-shape of the jaw plates, high stresses are experienced at the highest positions along the plate when coarser particles are fed and immediately get in contact with them. However, in order to simulate more precisely the behavior of the ore and conclude about the real stress patterns on the plates, it is necessary to take into account the decompression features of the PRM in the instant in time immediately after particle breakage. Such an improved description could be reached by an improved calibration of the relaxation parameter of the Particle Replacement Model, as presented by Barrios et al. [19].

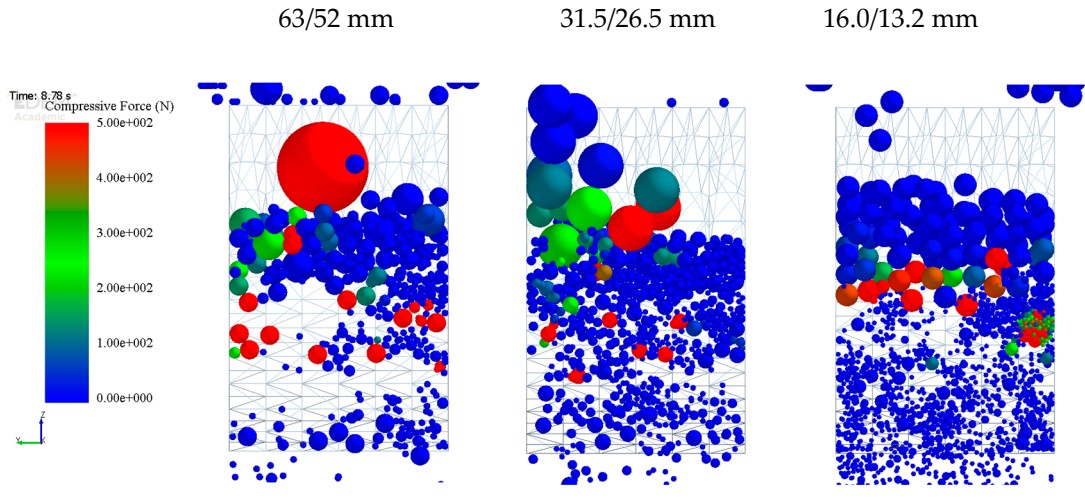

**Figure 10.** Particle bed compression inside the crushing chamber for different particle size classes fed to the crusher.

An equivalent analysis of the forces exerted on the plates can be seen from direct evaluation of the stress intensity on the plate geometry mesh (Figure 11). The figure is a snapshot of the DEM simulation that shows qualitatively the variation of the compressive force on the geometry of the swing and fixed plates. From the simulation it is observed that the area that is subjected to the highest stresses is located closest to the discharge opening, for all feed sizes. This is consistent with observations by Lindqvist and Evertsson [24], who studied the wear of the plates of a jaw crusher from both experiments and analytical models on the basis of measurement of pressure and forces exerted on the fixed and swing jaws. Figure 11 also shows that, when the crusher was fed with the coarser particles (63/52 mm), the plates were subjected to higher compressive forces.

Validation of the DEM simulation was carried out by comparing it to the experimentally measured values of throughput, power, and product size distribution. For instance, the mean net power demanded in experiments in which the crusher was fed with 63/52 mm material was 0.34 kW, whereas simulations yielded a mean value of 0.32 kW, thus demonstrating the very good agreement.

Figure 12 shows the variation of measured and the predicted net power of the jaw crusher, with good agreement between simulations and experiments. In the experiments, the highest powers were obtained when crushing the coarsest feed particles (63/52 mm). The fluctuation in the values were also found to be smaller for finer feeds. The peaks in power intensity represent the instants in which the machine squeezes and fractures either individual particles or assemblies of particles as they move down the crushing chamber, whereas the low values represent lack of particles being compressed by the crusher plates in those moments in time.

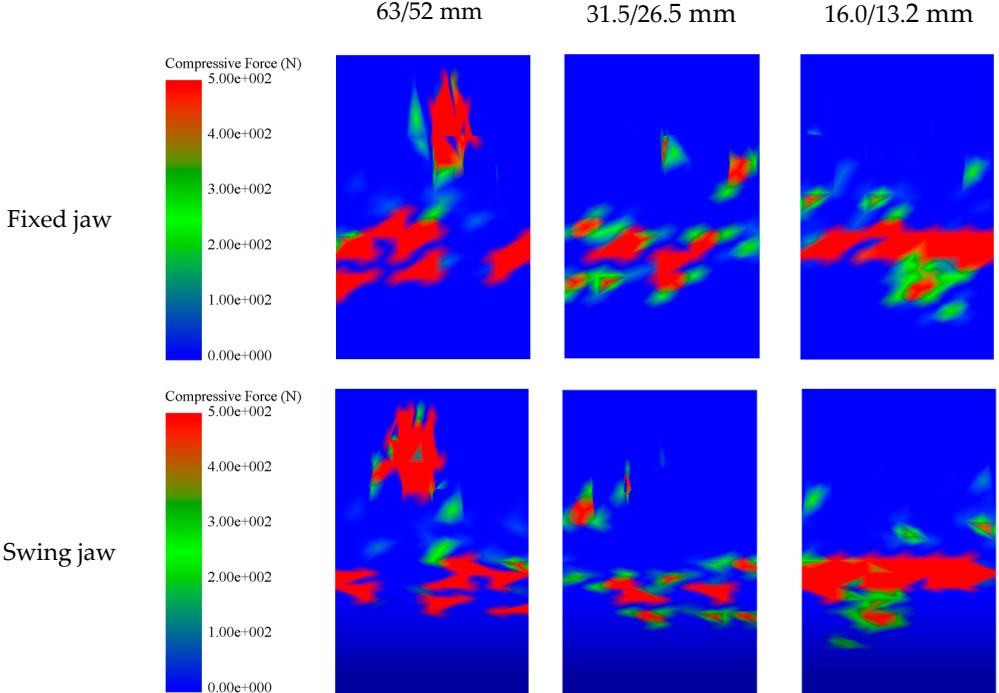

**Figure 11.** Snapshot of the compressive force on the geometries (fixed and swing jaws) for different feed particle sizes.

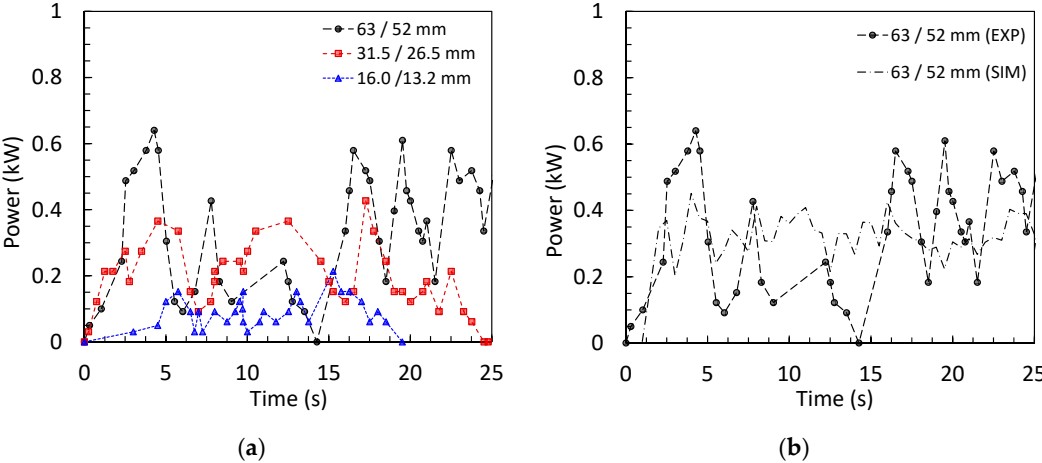

**Figure 12.** Measured (**a**) and comparison of measured and simulated (**b**) net power of the laboratory jaw crusher.

Figure 13 shows the variation of the throughput during the tests, which demonstrates that an increase in feed size resulted in a reduction in crusher throughput. The figure also shows the good agreement between the DEM simulation results and the measurements using the integrating load cell.

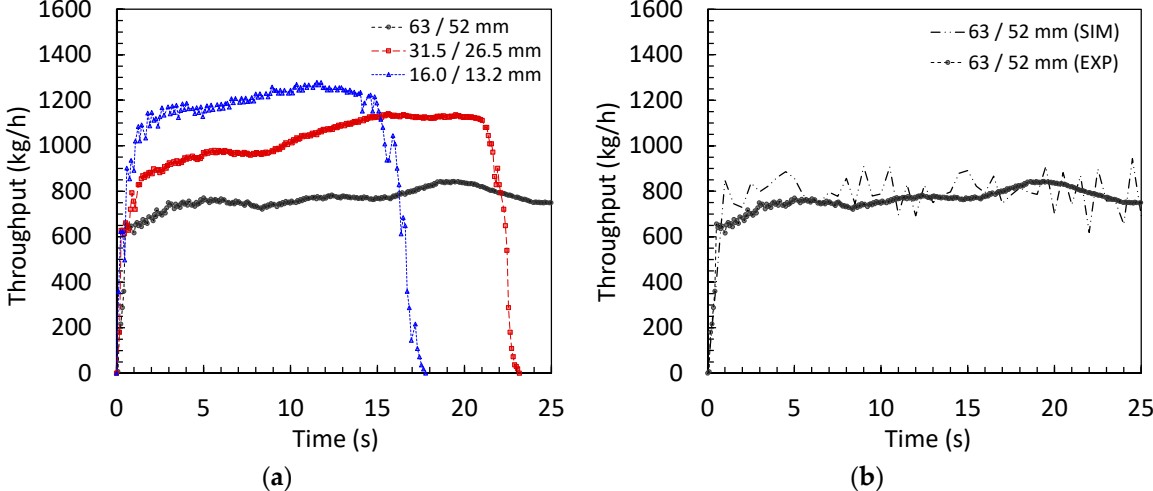

**Figure 13.** Experimental (**a**) and DEM simulations and experiment (**b**) of the laboratory jaw crusher throughput.

A more detailed comparison between experiments and simulation results is presented in Figure 14. The results show the good agreement between the experiments and the DEM simulations, in which the simulations captured very well the trends observed in the experiments regarding the effect of feed particle size on throughput and net power. Indeed, the reduction in throughput as particle size increased was well described by the DEM simulation (Figure 14a), as well as the increase in power required to crush the coarser feed particles (Figure 14b). In addition, Figure 14c compares measured and predicted product size distributions for the case in which the crusher was fed with a particle size distribution (Table 5). It shows very good agreement with the experimental results. Nevertheless, these are limited to the minimum sphere size simulated of 2.36 mm, given the additional computational cost associated to simulating finer particles.

Figure 14d shows the simulated compressive force for different feed particle sizes. The model shows that forces increase with feed particle size. This trend agrees with results from single particle experiments, as demonstrated by Tavares and King [25], as well as results from the present work (Figure 6). Unfortunately, it was not possible to validate this in the present work due to lack of proper instrumentation in the jaw crusher, such as load cells or others sensors, installed on the plates to register the changes in force, as carried out in earlier studies [24,26].

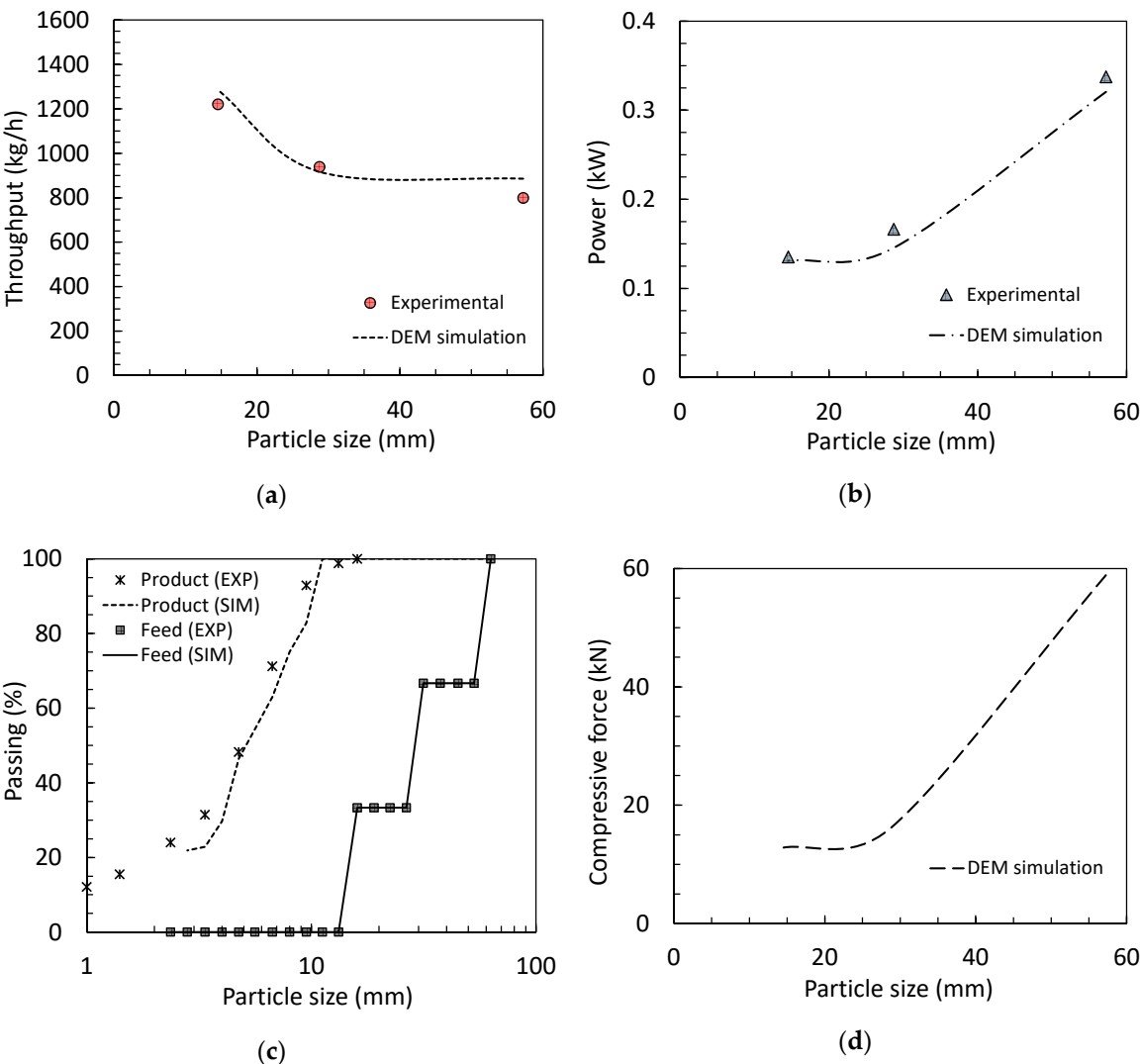

**Figure 14.** Comparison between experiments and DEM simulations for the jaw crusher throughput (**a**), power (**b**) and product size distribution (**c**); simulated trend on the compressive force (**d**). Figures a, b and d presented as a function of mean feed particle size.

### 3.3. Sensitivity Analysis of the Jaw Crusher DEM Model

Figure 15 shows the product particle size distributions for different stroke frequencies (Figure 15a) and closed side settings (Figure 15b) from the laboratory jaw crusher DEM simulations. Figure 15a shows a finer product for higher stroke frequencies for the same CCS, whereas Figure 15b shows a finer product for smaller CSS for a fixed value of frequency. A similar behavior of the product size distribution with frequency and closed side setting was found in earlier studies [6,14].

Time series showing the variation of the crusher throughput in the simulations as a function of frequency are presented in Figure 16. These data were extracted from the "particle mass flow sensor" of EDEM and filtered using a statistical moving mean (continuous line). The non-filtered data correspond to the markers. These data were extracted from the virtual experiments that considered a feed composed of multiple sizes operating in partially choke and non-choke feed condition during 7.5 s.

Results show the dynamics of the jaw crusher DEM model and the throughput response to the disturbances given by the stroke frequency of the swing plate. As observed in Figure 16, the DEM simulations respond to the changes in frequency. Additionally, more stable behavior with less scattered results from operation at the frequency of 9.0 Hz.

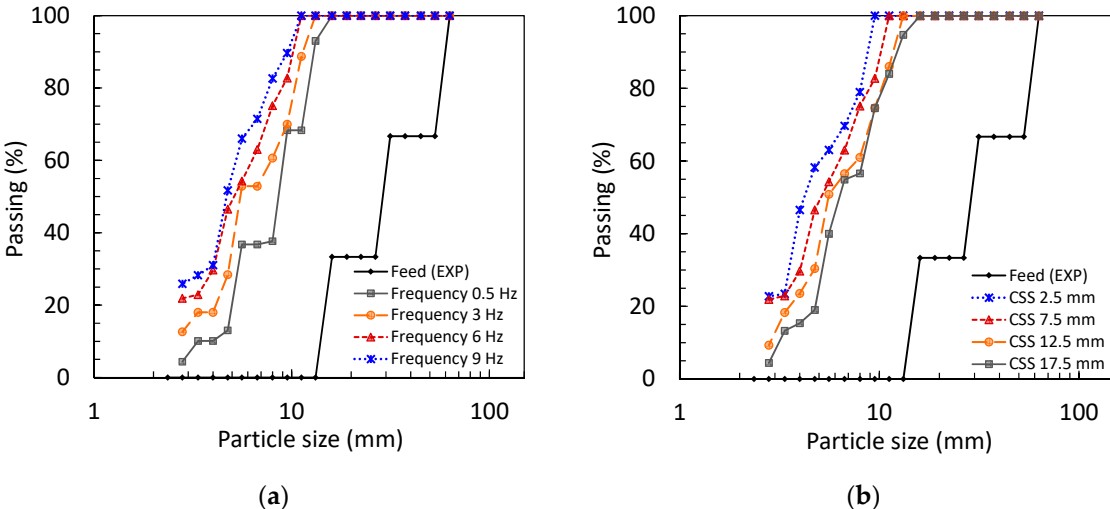

**Figure 15.** Product particle size distributions for different stroke frequencies and a constant CSS of 7.5 mm (**a**) and for different closed side settings for a fixed frequency of 6 Hz (**b**) in DEM simulations of the laboratory jaw crusher for a feed particle size distribution (Table 5).

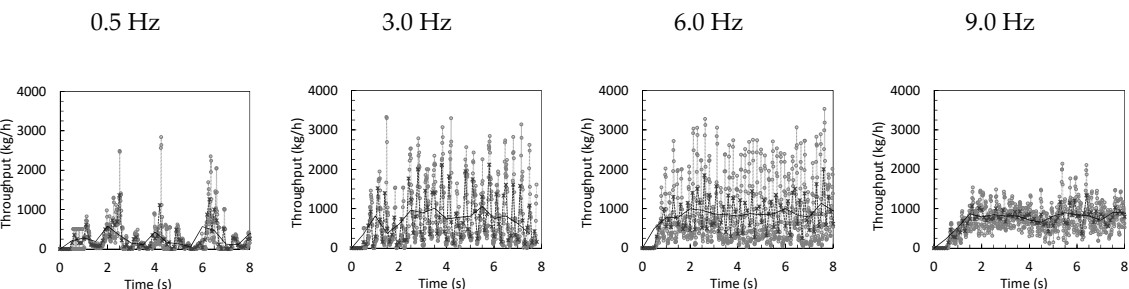

**Figure 16.** DEM jaw crusher simulation results showing raw throughput data for different stroke frequencies. Crusher fed with a range of sizes (63–13 mm) and CSS constant at 7.5 mm.

Figure 17 presents results on the sensitivity analyzes of the jaw crusher DEM model with the embedded Particle Replacement Model. It shows the variation of crusher capacity, compressive force, power and reduction ratio as a function of stroke frequency and closed side setting.

From Figure 17a, it is evident that throughput increases with closed side setting. This has been observed in several studies in the literature [6]. Regarding frequency, capacity initially increases, reaches a maximum value and then drops at higher frequencies. This trend is less pronounced for intermediate values of CSS, whereas it disappears at the largest values of CSS, in which throughput increases monotonically with frequency. As such, it is possible to conclude that high values of CSS combined with high frequencies apparently allow reaching the highest throughputs. The variation of throughput as a function of frequency was also described by Rose and English [10] using an analytical model, whereas the relationship among these variables has already been object of simulations by Johansson et al. [14] who found similar trends.

The effect of CSS and frequency on crusher power is shown in Figure 17b. It allows to conclude that power demanded by the jaw crusher increases significantly with frequency as well as with a reduction in closed side setting. A similar general trend was found elsewhere [14].

A further examination on the effectiveness of the crusher can be extracted from analyzing the reduction ratio ($R_{80}$). It expresses the ratio between the feed ($F_{80}$) and the product ($P_{80}$) 80% passing sizes of an operation [27]. Experimentally, it is typically desirable to reach the maximum values of reduction ratio for a given specific energy consumption. From the DEM model (Figure 17c) it is possible to conclude that the highest values of $R_{80}$ are reached both with the smallest CSS as well as the highest frequency. The same trend was also experimentally found by Fladvad and Onnela [6].

Figure 17d presents the resulting compressive force on the plates as a function of frequency and CSS. The highest compressive forces are achieved with smaller values of CSS. The results also show a modest increase in compressive forces with frequency.

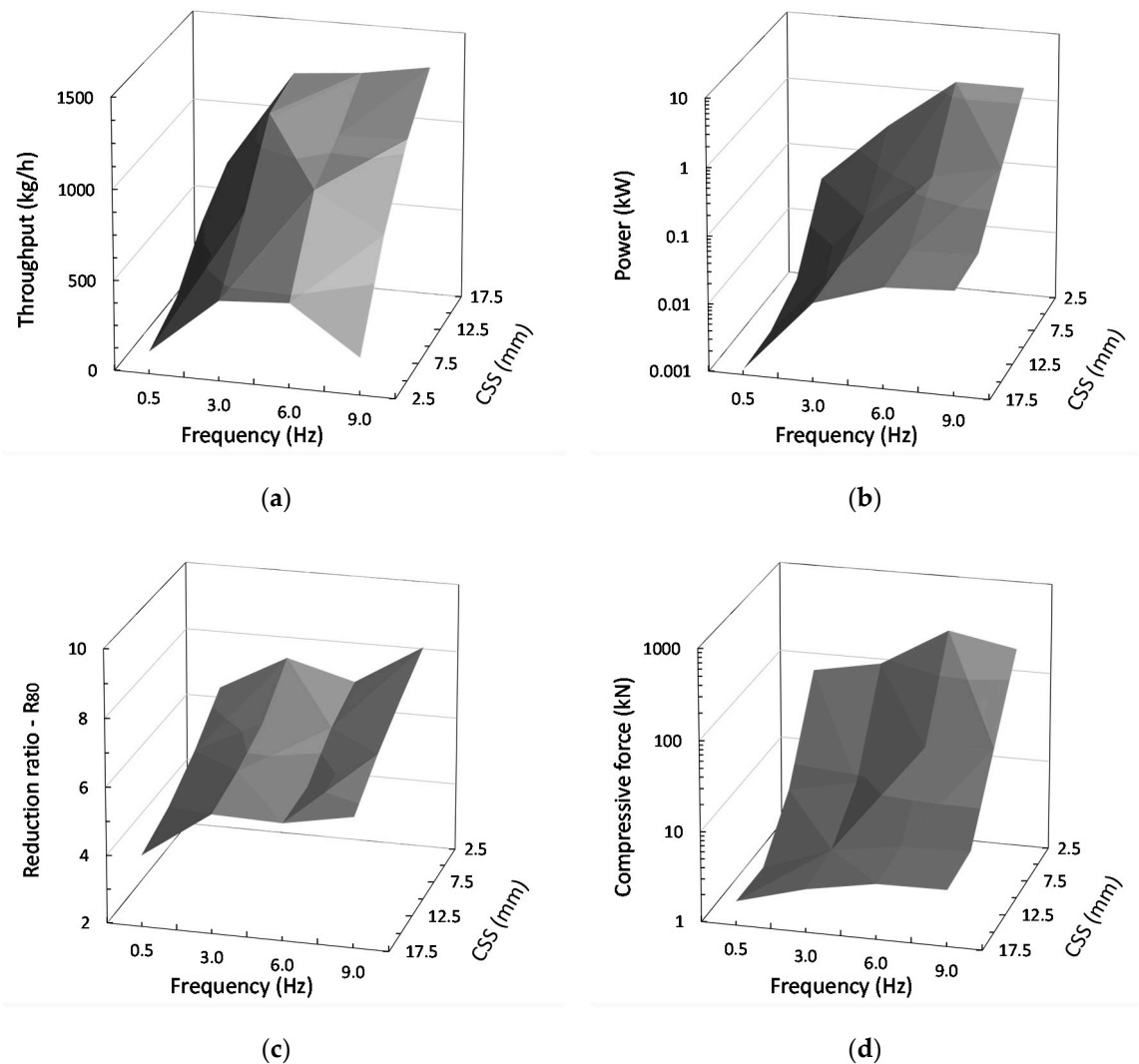

**Figure 17.** Response surface analysis of the jaw crusher DEM simulations showing throughput (**a**), power (**b**), reduction ratio (**c**) and compressive force (**d**) as a function of closed side setting (CSS) and stroke frequency.

## 4. Discussion

The DEM simulations results demonstrate the validity of properly calibrated simulations of the jaw crusher performance as a tool for improved design and optimization of industrial equipment. The approach of running simulations at a small (laboratory) scale allows the concept to be proved and the technology to be validated. The following step could be its application in the design of a customized jaw crusher with the aim of improving crusher performance. This could represent, for instance, setting the crusher frequency and stroke amplitude to values that maximize the machine performance for a given feed ore. Other potential applications include analyzes of jaw crusher operational response to fluctuations in particle size distribution in the feed, ore competence, as well as feed segregation. On the other hand, such a tool could be used by equipment manufacturers to customize the machine design in terms of jaw wear surface and crusher chamber geometry.

In particular, the use of PRM is an efficient solution to represent, in a realistic way, the performance of the jaw crusher in terms of throughput, specific energy consumption, compressive force, and product particle size distribution. This model offers some computational benefits in comparison with other DEM breakage methods, such as the Bonded-Particle Model [17]. Among its greatest advantages is its reduced computational effort, ability to describe different loading conditions, ease in model parameter estimation and calibration, besides reasonably good resolution in describing the product size down to relatively fine sizes.

## 5. Conclusions

Simulations of a laboratory jaw crusher with a Particle Replacement Model embedded in DEM were able to reproduce the experimental performance of a laboratory jaw crusher in size reduction of a gold ore from the department of Huila–Colombia, in terms of power, throughput, and product size distribution. Very good agreement was observed between measured and predicted results obtained for different feed sizes.

The DEM model demonstrated high sensitivity to changes in CSS and swing jaw frequency, allowing to show some trends in throughput, power, reduction ratio, and compressive force. The trends observed were generally in good agreement with the literature.

Simulations of the jaw crusher incorporating a description of particle breakage can be a useful tool in design and optimization of the machine, since DEM may be an excellent tool to predict machine performance, including assessing measures that are hard to obtain experimentally, such as the stress gradient on the jaw plates.

**Author Contributions:** Conceptualization, G.K.B. and N.J.-H.; methodology, G.K.B.; software, G.K.B.; validation, G.B., N.J.-H. and S.N.F.-T.; formal analysis, G.K.B. and N.J.-H.; investigation, G.K.B. and N.J.-H.; resources, G.K.B., N.J.-H. and L.M.T.; data curation, G.K.B.; writing—original draft preparation, G.K.B., N.J.-H., S.N.F.-T. and L.M.T.; writing—review and editing, G.K.B., N.J.-H. and L.M.T.; visualization, G.K.B. and N.J.-H.; supervision, G.K.B. and N.J.-H.; project administration, G.K.B. and L.M.T.; funding acquisition, G.K.B., N.J.-H., S.N.F.-T. and L.M.T. All authors have read and agreed to the published version of the manuscript.

**Funding:** One of the authors (L.M.T.) would also like to thank funding from the Brazilian Agencies CNPq (grant number 310293/2017-0) and FAPERJ (grant number E-26/202.574/2019).

**Acknowledgments:** The authors wish to thank the Servicio Geológico Colombiano, based on Cali for the support by means of its infrastructure, equipment and laboratories. The authors also thank the EDEM Company for providing the EDEM software by means of "Take EDEM with you" program.

**Conflicts of Interest:** The authors declare no conflict of interest.

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
