# Peer review of "DEM Simulation of Laboratory-Scale Jaw Crushing of a Gold-Bearing Ore Using a Particle Replacement Model"

_minerals, doi:10.3390/min10080717_

Round 1

Reviewer 1 Report

The paper presents a validation study of a DEM breakage model using experimental data obtained for a Jaw crusher. The calibration of the model uses a single particle compression case and adjustment of progeny size distribution for each breakage event based on laboratory jaw crusher exits. The calibrated model shows good agreement to expt data for a number of measurements using the real world device.  

I would like to see a more comprehensive comparison of the product PSD from the simulation and the expt. Fig. 14 (C) displays this, but it is crushed vertically due to the x-axis range used. A detailed comparison for the different size classes and comments on where the model is over/under representing components of the PSD would give very useful information on model performance and potential areas for further improvements. 

Author Response

Reviewer 1

Comments and Suggestions for Authors

The paper presents a validation study of a DEM breakage model using experimental data obtained for a Jaw crusher. The calibration of the model uses a single particle compression case and adjustment of progeny size distribution for each breakage event based on laboratory jaw crusher exits. The calibrated model shows good agreement to expt data for a number of measurements using the real world device. 

I would like to see a more comprehensive comparison of the product PSD from the simulation and the expt. Fig. 14 (C) displays this, but it is crushed vertically due to the x-axis range used.

Response: Thanks for the suggestion. The Authors changed the x axis of the Figure 14 (c) in order to allow a more comprehensive comparison of the experimental and DEM simulated PSD.

A detailed comparison for the different size classes and comments on where the model is over/under representing components of the PSD would give very useful information on model performance and potential areas for further improvements.

Response: The Authors included an explanation of the PSD result in the line 252, showing the challenges of the model to represent small sphere sizes. In addition the Authors included the “Discussion” section, on it, the practical benefits and the potential areas of application of the PRM are discussed.

Reviewer 2 Report

The paper concerns problems of modeling and simulation of jaw crusher performance. Article is written in rather good style, minor language corrections can be done. Experimental programme and the obtained results are described adequately, however additional comments on potential application of the findings of paper should be added (see comments below).

The main concern is that both the DEM method and the jaw crusher itself are widely described in literature and many research over this issue were already presented. Thus there should be added in the paper what is an added value of these investigations. The PRM method should be described in more details as the readers not familiar with this type of simulation might face problems with understanding the idea. The structure of manuscript should be modified and the “Discussion” section has to be added. The practical benefits and their utilization should be commented in this section.

Detailed remarks:

  • Please modify the title of paper and add information on the material type examined, i.e. “DEM simulation (…) using of PRM model on the example of gold-bearing ore”;
  • References should be supplemented, esspecially in the section where the authors describe investigations on jaw crusher performance (lines 39-50). Works of Gawenda, Naziemiec and Tumidajski concerning modeling of jaw crushers operation could be mentioned;
  • Please provide the PSD of feed material instead of Figure 1;
  • Remove Figs 2 and 3, as they demonstrate a general knowledge and do not point any new aspect. Instead of this an idea or scheme of investigative programme is welcome.

Language/editorial comments:

  • Line 33 and 45: replace “gape” by “gap”;
  • 33 and 36: use “material” instead of “ore” because it is a general description of the device;
  • Line 37: replace “smaller” by “finer”;
  • Lines 37-38: the second part of the sentence sounds weird, please modify it;
  • Line 82: replace “grain” by “particle”;
  • 248-249: reformulate the sentence into: “Models show that forces increase together with increasing of feed particle size"

Author Response

Reviewer 2

Comments and Suggestions for Authors

The paper concerns problems of modeling and simulation of jaw crusher performance. Article is written in rather good style, minor language corrections can be done. Experimental programme and the obtained results are described adequately, however additional comments on potential application of the findings of paper should be added (see comments below).

The main concern is that both the DEM method and the jaw crusher itself are widely described in literature and many research over this issue were already presented. Thus there should be added in the paper what is an added value of these investigations. The PRM method should be described in more details as the readers not familiar with this type of simulation might face problems with understanding the idea.

Response: Although the jaw crusher has been the object of some publication, the application of the PRM, which is a very computationally-efficient method, reaching the good fidelity that we have, is certainly not commonplace in the literature. Responding the reviewer, the Authors included a description of the PRM and the related references where the model used in this research is presented in detail.

The structure of manuscript should be modified and the “Discussion” section has to be added. The practical benefits and their utilization should be commented in this section.

Response: Following the reviewer´s suggestion, the Authors included the “Discussion” section. On it, the practical benefits and the potential areas of application of the PRM are discussed.

Detailed remarks:

Please modify the title of paper and add information on the material type examined, i.e. “DEM simulation (…) using of PRM model on the example of gold-bearing ore”;

Response: The Authors modified the paper title according to the reviewer suggestion.

References should be supplemented, esspecially in the section where the authors describe investigations on jaw crusher performance (lines 39-50). Works of Gawenda, Naziemiec and Tumidajski concerning modeling of jaw crushers operation could be mentioned;

Response: The authors included the suggested reference in the introduction section, adding value to the literature review.

Please provide the PSD of feed material instead of Figure 1;

Response: Figure 1 shows the Iquira Huila gold-bearing ore stockpile at the pilot plant of the Colombian Geological Survey. The Run of Mine Sample has around 3 tonnes of gold-bearing ore with particles in a wide range of sizes between 90 mm and fines under 0.1 mm. The detailed PSD of the ROM was not described in the paper, the jaw crusher was fed with different particle narrow sizes and a distribution of particles, in order to better describe the feeding particle size effect on the Jaw crusher performance.

Remove Figs 2 and 3, as they demonstrate a general knowledge and do not point any new aspect. Instead of this an idea or scheme of investigative programme is welcome.

Response: The authors consider the Figure 2, which shows the laboratory jaw crusher machine used in the present work, is important so that the reader can appreciate exactly the scale and main features of the machine, whereas many of the key variables cited along the manuscript are explained in detail in Figure 3. I hope the reviewer can understand that, for these reasons, the Authors insist in keeping both figures.

Language/editorial comments:

Line 33 and 45: replace “gape” by “gap”;

Response: Instead, the word “gape” was replaced by top opening.

33 and 36: use “material” instead of “ore” because it is a general description of the device;

Response: The word “material” was replaced by “ore”.

Line 37: replace “smaller” by “finer”;

Response: The word “smaller” was replaced by “finer”.

Lines 37-38: the second part of the sentence sounds weird, please modify it;

Response: The sentence was corrected.

Line 82: replace “grain” by “particle”;

Response: The word “grain” is related to the specific mineral species. For example, a gold-bearing ore particle is composed of many mineral species and gold. The gold particles included in an ore particle are denominated “grains”. As such, it would not be appropriate to follow the reviewer´s suggestion.

248-249: reformulate the sentence into: “Models show that forces increase together with increasing of feed particle size";

Response: The sentence was reformulated according the reviewer suggestion.